# Argument-based Detection and Classification of Fallacies in Political Debates

**Pierpaolo Goffredo**[1], **Mariana Chaves Espinoza**[1], **Elena Cabrio**[1], **Serena Villata**[1]

[1] Université Côte d'Azur, CNRS, Inria, I3S, France

goffredo@i3s.unice.fr, machaves@i3s.unice.fr,
elena.cabrio@univ-cotedazur.fr, serena.villata@univ-cotedazur.fr

## Abstract

Fallacies are arguments that employ faulty reasoning. Given their persuasive and seemingly valid nature, fallacious arguments are often used in political debates. Employing these misleading arguments in politics can have detrimental consequences for society, since they can lead to inaccurate conclusions and invalid inferences from the public opinion and the policymakers. Automatically detecting and classifying fallacious arguments represents therefore a crucial challenge to limit the spread of misleading or manipulative claims and promote a more informed and healthier political discourse. Our contribution to address this challenging task is twofold. First, we extend the ElecDeb60To16 dataset of U.S. presidential debates annotated with fallacious arguments, by incorporating the most recent Trump-Biden presidential debate. We include updated token-level annotations, incorporating argumentative components (i.e., claims and premises), the relations between these components (i.e., support and attack), and six categories of fallacious arguments (i.e., Ad Hominem, Appeal to Authority, Appeal to Emotion, False Cause, Slippery Slope, and Slogans). Second, we perform the twofold task of fallacious argument detection and classification by defining neural network architectures based on Transformers models, combining text, argumentative features, and engineered features. Our results show the advantages of complementing transformer-generated text representations with non-textual features.

## 1 Introduction

Fallacious arguments have been firstly defined as defective inferences, i.e., logically invalid types of arguments (Eemeren, 2001). More recently, in a more pragmatic perspective, fallacious arguments have been defined as infringements of performance rules characteristic of a particular ideal type of argumentative engagement (Eemeren and Grootendorst, 1987) and as illicit dialectical shifts across different dialogue types, highlighting that the attempted move is inappropriate with respect to its pragmatic application context (Walton, 1995). Despite their employment in many scenarios (e.g., online discussion platforms and blogs, TV roundtables), a natural testbed of this form of misleading argumentation is political debate. For instance, the *ad hominem* fallacy, where the plausibility of the argument depends on the credentials, personal background and past actions of the speaker, is probably one of the fallacy labels that is most often thrown around in political debate. This kind of arguments may sound convincing and has the goal to mislead the audience, persuading it about the validity of the argument. Given the potential nefarious impact of these misleading arguments on the society, identifying and classifying fallacious arguments is therefore a main open challenge in Argument Mining (AM) (Cabrio and Villata, 2018; Lawrence and Reed, 2019; Lauscher et al., 2022), and in NLP in general.

Existing approaches in the literature (Habernal et al., 2017, 2018; Goffredo et al., 2022; Vijayaraghavan and Vosoughi, 2022; Alhindi et al., 2022; Vorakitphan et al., 2022; Sahai et al., 2021) mainly concentrated on the classification of fallacious text snippets over a finite set of labels, leaving the challenging issue of identifying the fallacious text snippet and its boundaries under-investigated. In this paper, we tackle this open research question on a dataset of political debates from the U.S. presidential campaigns from 1960 to 2020.

More precisely, the contribution of this paper is twofold. First, we extend an existing resource of U.S. political debates from the presidential campaigns (1960-2016) annotated both with argument components and relations, and six fallacy categories (namely, *Ad Hominem, Appeal to Authority, Appeal to Emotion, False Cause, Slippery Slope and Slogans*). This new resource, named the ElecDeb60to20 dataset, includes now also the debates of the 2020 presidential campaign (Trump-

Biden) with all the related annotations. Second, we propose a new approach, based on Transformers, to detect the fallacious text snippets in these debates, and then classify them along the six fallacy categories. This approach encodes the argument components (i.e., *Premise, Claim*), the argument relations (i.e., *Support, Attack*) and the PoS tags to successfully identify and classify fallacious arguments. Experimental results show that the proposed approach outperforms standard baselines and concurrent approaches with an average f1-score of 0.74, with our proposed model named MultiFusion BERT, on the task of fallacious argument detection and classification.

Whilst most of the computational approaches targeting fallacious argumentation focus on the pure classification of such nefarious content (Habernal et al., 2017, 2018; Jin et al., 2022; Goffredo et al., 2022; Vijayaraghavan and Vosoughi, 2022; Alhindi et al., 2022), the originality of our contribution is that it proposes, to the best of our knowledge, the first neural architecture to both detect fallacious arguments and classify them in political debates, and one of the very few approaches to tackle this task in general, outperforming competing approaches (Vorakitphan et al., 2022; Sahai et al., 2021).

The urgency to study fallacies in political discourse is crucial both from the philosophical and the political perspective. It emphasizes the need to scrutinize political arguments for sound reasoning rather than deception. Understanding logical flaws is a main issue for informed decision-making, enabling the recognition and assessment of fallacious arguments in political discourse. Moreover, examining how fallacies are employed in political debates reveals their strategic role in influencing opinion and diverting attention. This strategic use of fallacies mirrors the subtleties of language, emphasizing the interplay between rhetoric, philosophy, and political communication. Furthermore, it offers valuable insights into the dynamics of public discourse and the need for critical analysis (Walton, 1987, 1995).

## 2 Related Work

Over the years, there has been a growing interest in the field of NLP to the detection of fallacies and related phenomena, including misinformation and propaganda (Da San Martino et al., 2020b). The pioneering work of Da San Martino et al. (2019b) on fallacies in newspaper news has been a signif-

icant source of inspiration in this area. Recently, researchers have made significant progress in identifying and classifying fallacies within discourse. In this section, we discuss the approaches proposed in the literature on the two tasks of fallacy classification (Section 2.1) and fallacy detection (Section 2.2).

### 2.1 Fallacy Classification

For instance, Habernal et al. (2017) aimed to improve fallacy detection by creating a publicly available software called "Argotario". It serves as an educational gaming platform and a means to gather data from the crowd for annotating fallacy types in everyday arguments. In a subsequent study, Habernal et al., 2018 released annotated datasets of fallacious arguments in English and German. They conducted experiments using Support Vector Machine and BiLSTM models with German word vectors to classify six fallacious topic types (accuracy=50.9%, macro-F1 score=42.1%).

Jin et al. (2022) proposes an architecture based on a simple classifier that incorporates the structural information of fallacies. They augment a standard pre-trained language model for classification with a template that includes modified assumptions and premises based on a well-defined masking scheme. They achieve an F1 score of 58.77% on the classification task over 13 classes. They used a dataset collected from online teaching materials, specifically designed to teach and test students' understanding of logical fallacies.

In a more contextual approach, Goffredo et al. (2022) use a dataset from U.S. Presidential Election Debates From 1960 to 2016. This dataset incorporates information such as the context and argumentative features (i.e., argument components and relations) for each fallacy. The proposed architecture uses one classifier for each feature in order to calculate the loss. On a sentence classification task encompassing six fallacy categories, their approach achieves an F1 score of 84%.

To address the identification and classification of fine-grained propaganda tweets, Vijayaraghavan and Vosoughi (2022) proposed an end-to-end transformers-based approach that considers additional features like context, relational information, and external knowledge through data augmentation. Their dataset consists of approximately 211K tweets annotated with 19 classes (18 propaganda types and 1 non-propaganda). The approach

yielded a 64% F1 score on a text classification task.

Alhindi et al. (2022) introduced an instruction-based prompt in a multitask configuration using the T5 model to classify fallacies. This methodology involved leveraging multiple fallacy-based datasets, including Propaganda (Da San Martino et al., 2019b), Logic (Jin et al., 2022), Argotario (Habernal et al., 2017), Covid-19 (Musi et al., 2022), and Climate (Jin et al., 2022). Their approach enabled the identification of 28 distinct fallacies across various domains and genres, facilitating the analysis of model size and prompt selection, and investigating the impact of annotation quality on model performance, potentially supplemented with external knowledge. The results obtained using their T5-large model yielded F1 scores of 41% for Propaganda, 62% for Logic, 59% for Argotario, 26% for Covid-19, and 17% for Climate, respectively.

## 2.2 Fallacy Detection and Classification Task

Vorakitphan et al. (2022) proposes a system capable of automatically identifying propaganda messages and classifying them based on the propaganda techniques employed. The system adopts a pipeline approach that firstly detects the text snippet containing potential propaganda, and then performs classification by leveraging semantic and argumentative features. Two standard benchmarks, NLP4IF'19 (Da San Martino et al., 2019a) and SemEval'20 datasets (Da San Martino et al., 2020a), were used for this propaganda detection and classification task. The binary classification step, utilizing BERT, achieved a 72% F1 score, while the sentence-span multi-class classification task (14 classes) achieved a 64% micro F1 score using a RoBERTa-based architecture.

Fallacy detection and token-level classification tasks were also performed in Sahai et al. (2021). They narrowed their study to 8 fallacy types and created a corpus of fallacious arguments by annotating user comments on Reddit. They performed fallacy classification at comment-level and token-level, relying on BERT and MGN (Da San Martino et al., 2019a) models. The inclusion of the conversation context, represented by the parent comment or submission title, was used to enhance the predictions. Fine-tuned BERT with a classification head of a linear layer reported the best results on the token classification task (macro F1=53%).

Our present work expands the scope of these

fallacy studies in several ways. Firstly, we employ a corpus specifically created for fallacy detection in political debates, which has been under investigated in this context even though fallacious argumentation is a main issue in political discourse. Secondly, while the majority of existing studies focus on fallacy classification, in this paper we focus on the fallacy detection task, which is of main importance in real-life scenarios where text or speech lacks pre-segmentation and a clear binary classification into fallacies or non-fallacies. Lastly, our study capitalizes on the annotations of argumentation features in our dataset, an element that is often absent in other corpora, allowing us to go beyond pure text-based approaches, and to rely on the argumentation structure.

## 3 ElecDeb60to20 Dataset

To effectively address the task of detecting and classifying fallacious arguments within political debates, we decided to rely on the ElecDeb60To16 dataset (Haddadan et al., 2019; Goffredo et al., 2022). It comprises televised debates from U.S. presidential election campaigns spanning from 1960 to 2016. These debates were sourced from the website of the Commission on Presidential Debates[1], which openly provides transcripts of debates broadcasted on television and featuring the prominent candidates for presidential and vice-presidential nominations in the United States. All information on this website is accessible to the public. Considering the most recent presidential election between Trump and Biden occurred in 2020, we expanded the dataset with the transcripts of the debates of this election campaign to include updated annotations, incorporating argumentative components such as *Claims* and *Premises*, as well as the relations between these components, i.e., *Support* and *Attack*. As a result of this annotation update, the dataset is renamed as ElecDeb60to20[2], reflecting the coverage of debates spanning from 1960 to 2020. This updated dataset provides a more comprehensive and contemporary collection of fallacies in political debates, enhancing the relevance and applicability of the data for further analysis and research in the field.

This resource is a valuable benchmark for investigating potential connections between specific

---

[1] https://www.debates.org/voter-education/debate-transcripts/
[2] https://github.com/pierpaologoffredo/FallacyDetection

argument components and relations that underlie the occurrence of fallacious arguments.

## 3.1 Annotated Fallacies

During the annotation process of fallacies within the U.S. political debates of the 2020 presidential election, we rely on the six categories based on the annotation scheme proposed by Da San Martino et al. (2019a), the categorization outlined by Walton (1987), and the annotation of the previous debates in the first version of the dataset with these fallacy categories (Goffredo et al., 2022). Hence, we adopted the following categories: *Ad Hominem*, *Appeal to Authority*, *Appeal to Emotion*, *False Cause*, *Slippery Slope*, and *Slogans*. Below, we provide a concise description of each of these six categories.

**Ad Hominem.** When the argument becomes an excessive attack on an arguer's position (Walton, 1987).

**Appeal to Emotion.** The unessential loading of the argument with emotional language to exploit the audience emotional instinct.

**Appeal to Authority.** It occurs when the arguer relies on the endorsement of an authority figure or a group consensus without providing sufficient evidence. It may also involve the citation of non-experts or the majority to support their claim.

**Slippery Slope.** This fallacy implies that an improbable or exaggerated consequence could result from a particular action.

**False Cause.** The misinterpretation of the correlation of two events for causation (Walton, 1987).

**Slogan.** It is a brief and striking phrase used to provoke excitement of the audience, and is often accompanied by another type of fallacy called *argument by repetition*.

## 3.2 Annotation Phase

The updated annotations were conducted following the annotation scheme introduced in Haddadan et al. (2019); Goffredo et al. (2022). Following this approach, each debate was divided into sections, starting with either a moderator/panelist or an audience member asking a question on a new topic. To facilitate the annotation process, the semantic annotation platform INCEpTION (Klie et al., 2018) was used.

Two annotators, with expertise in computational linguistics, independently annotated the new portion of the dataset (Trump vs. Biden debates) by identifying argumentative components, relations, and fallacies. To maintain objectivity and prevent bias, the annotation process for argumentative components was performed on raw data, without any pre-existing fallacy annotations. This approach was adopted to ensure that the annotation process remains unbiased and free from any preconceived notions related to fallacies. A set of 50 sentences randomly extracted from the debates was annotated to assess Inter-Annotator Agreement (IAA), and the results, visualized in Table 1, indicate a substantial level of agreement between the annotators.

| Measure | Value |
|---------|-------|
| Observed Agreement | 0.857 |
| Krippendorff's $\alpha$ | 0.757 |

Table 1: IAA agreement over 50 sentences randomly extracted from the debates Trump-Biden.

## 3.3 Statistics and Data Analysis

Table 2 summarizes the Trump vs. Biden's debates annotations per category and argumentative features. We tokenized the annotated fallacious arguments to compute the average number of words in each category. In line with the guidelines of Goffredo et al. (2022), *Slogans* is the shortest with 5.0 tokens on average[3], whereas *SlipperySlope* was the longest with 20.5 tokens on average.

| Category | Freq | AvgTok | Arg. Feature | Freq |
|----------|------|--------|--------------|------|
| Ad Hominem | 62 | 4,6 | Claims | 1513 |
| AppealtoAuthority | 17 | 18,6 | Premise | 332 |
| AppealtoEmotion | 147 | 6,81 | Support Rel. | 400 |
| FalseCause | 0 | 0 | Attack Rel. | 112 |
| SlipperySlope | 4 | 20,5 | | |
| Slogans | 2 | 5 | | |
| **Total** | **232** | **9,25** | **Total** | **2357** |

Table 2: Distribution of annotated fallacies per category and argumentative features of Trump vs. Biden's debates.

The train and test set split was performed considering the entire new dataset ElecDeb60to20. The training set accounts for 90% of the dataset, while the remaining 10% constitutes the test set. The distribution of fallacy labels is as follows: *AppealtoEmotion* (59.94%), *AppealtoAuthority* (15.20%),

---

[3] *Slogan* is by definition a short and striking phrase.

*AdHominem* (13.58%), *FalseCause* (46.93%), *Slipperyslope* (3.97%), and *Slogans* (2.63%). In the last debate, the most used fallacies are *AppealtoEmotion* and *AdHominem*, confirming the trend of the previous debates. Behind this strategy, there are many references to the COVID-19 pandemic and some personal issues of the two candidates exploited during the debates. Despite being distinct and unrelated, these two topics held significant importance and consistently fueled intense debates.

## 4 Fallacy Detection

We cast the fallacy detection task as an information extraction problem, where the goal is to identify and classify in the debates the textual snippets corresponding to the six categories of fallacies annotated in the context of a political debate (see Section 3.1 for the list of fallacies and their description).

We rely on the *BIO/IOB* data format, and specific tags are assigned to annotate the fallacies, i.e., *B-AdHominem*, *I-AdHominem*, *B-AppealtoAuthority*, *I-AppealtoAuthority*, *B-AppealtoEmotion*, *I-AppealtoEmotion*, *B-FalseCause*, *I-FalseCause*, *B-Slipperyslope*, *I-Slipperyslope*, *B-Slogans*, *I-Slogans*, *O*.

The fallacy detection and classification tasks consist therefore in assigning one of these thirteen predefined labels to each token.

To have a richer representation of fallacy annotations, we build a contextual framework that includes the sentence containing the fallacy, as well as the *preceding* and *following* sentences. When the fallacious sentence is the first or last in the dialogue, the preceding or following sentence is excluded.

### 4.1 Method

To address the above-mentioned tasks, we employ transformer-based architectures in both their basic configuration and in a specialized configuration designed for token classification[4], drawing inspiration from previous studies on fallacy detection and classification (Da San Martino et al., 2020a; Vorakitphan et al., 2022; Goffredo et al., 2022) that have provided empirical evidence of the advantages of complementing transformer-generated text representations with non-textual features. Moreover, we enhance the specialized architecture by including additional argumentative features.

---

[4]https://huggingface.co/docs/transformers/tasks/token_classification

### 4.1.1 Baselines

**BERT + (Bi)LSTM(s)**  The simplest models consist of a pre-trained BERT model followed by either *(i)* an LSTM layer and a dense layer, or *(ii)* a BiLSTM layer with 0.2 dropout, an LSTM layer, and a dense layer. The weights of the transformer are kept frozen during training. The text serves as input for the transformer, and we extract the last hidden states (i.e., the embedded representation of each token). This output is then passed on to the subsequent layers. In the case where argumentative features are included in the model, we concatenate the last hidden states of the transformer with the one-hot-encoded representation of the argument components and relationships. This concatenated feature representation is then fed into the next RNN-based layers. All models used Adam optimizer with default PyTorch parameters.

**BertForTokenClassification**  is a transformer-based model relying on a bidirectional approach to capture contextual information from surrounding words. We tested two checkpoints: `bert base uncased` and `bert-large-cased-finetuned-conll03-engl`.

**DebertaForTokenClassification**  is based on a modified transformer architecture with improvements like "de-coupled attention" and "cross-layer parameter sharing" for enhanced language modeling capabilities. The checkpoint used is `microsoft/deberta-base`.

**ElectraForTokenClassification**  relies on a novel pre-training method called "discriminative pre-training," where a generator and a discriminator are trained to enhance the quality of the learned representations. The checkpoint used is: `bhadresh-savani/electra-base-discriminator-finetuned-conll03-english`.

**DistilbertForTokenClassification**  is a distilled version of BERT that retains much of its performance while significantly reducing the model size and computational resources required for training and inference. The checkpoint used are: `distilbert-base-cased` and `distilbert-base-uncased`.

### 4.1.2 MultiFusion BERT

Relying on the results obtained by the baselines on the fallacy detection task on

| Model | Avg macro F1 Score |
|---|---|
| BERT + LSTM | 0.4697 |
| BERT + LSTM (comp. and rel. features) | 0.5142 |
| BERT + BiLSTM + LSTM | 0.5495 |
| BERT + BiLSTM + LSTM (comp. and rel. features) | 0.5614 |
| BertFTC `bert-base-uncased` | 0.7096 |
| BertFTC `dbmdz/bert-large-cased-finetuned-conll03-english` | **0.7237** |
| DebertaFTC `microsoft/deberta-base` | 0.7222 |
| ElectraFTC `bhadresh-savani/electra-base-discriminator-finetuned-conll03-english` | 0.4033 |
| DistilbertFTC `distilbert-base-cased` | 0.7010 |
| DistilbertFTC `distilbert-base-uncased` | 0.7047 |
| MultiFusion BERT (comp., rel. and PoS features) | **0.7394** |

Table 3: Average macro F1 scores for fallacy detection (BIO labels are merged) using different models. The scores are based on an average of 3 runs, except for BERT + (Bi)LSTM(s) models, which were evaluated using 10 runs. (FTC stands for "ForTokenClassification)

| Features | | | Avg macro |
|---|---|---|---|
| Components | Relationships | PoS | F1 Score |
| ✓ | | | 0.6922 |
| | ✓ | | 0.6922 |
| | | ✓ | 0.7212 |
| ✓ | ✓ | | 0.7278 |
| ✓ | | ✓ | 0.7166 |
| | ✓ | ✓ | 0.7166 |
| ✓ | ✓ | ✓ | **0.7394** |

Table 4: Average macro F1 scores for fallacy detection (BIO labels are merged) using MultiFusion BERT and different features. The scores are based on an average of 3 runs.

the ElecDeb60to20 dataset reported in Table 3[5], we select the BertFTC model (`bert-large-cased-finetuned-conll03-eng.`) to be included in our proposed architecture.

Despite the good performances of BertFTC, we propose to enhance its capabilities to detect fallacious text by integrating and "fusing" additional features, namely argumentative components (*Claim, Premise*), argumentative relations (*Support, Attack*), and Part-of-Speech (PoS) tags.

Argumentative features were included to improve the model's understanding of an argument underlying structure, enabling it to detect when its logical structure is compromised. Since fallacies involve faulty reasoning, we hypothesized that providing this information to the model would be relevant. Results in Table 4 provide empirical evidence to support this hypothesis. In particular, it has been shown that including argumentation features in-

creases the model performances in the context of fallacy classification (Goffredo et al., 2022). Such features can be extracted by specific annotations associated with each fallacy in the ElecDeb60to20 dataset: the type of argumentative component in which the fallacy may be present, and the argumentative relations among these different components.

The integration of PoS information is driven by the observation that certain fallacies exhibit distinctive language patterns that can be more easily discerned using PoS tagging. For example, in the *LoadedLanguage* fallacy (a subcategory of the *AppealToEmotion* category), the intensity of a sentence is often increased by using emotionally loaded phrases, expressed through the use of a sentiment lexicon, particularly concerning adjectives and adverbs. Similarly, in the *AdHominem* fallacy, where the focus shifts from attacking the argument to targeting the character, motives, or personal qualities of the political opponent, the reference to this opponent is expressed using a noun or pronoun and subsequently employs adjectives with negative connotations.

Figure 1 illustrates the proposed model, called MultiFusion BERT, for the detection and classification of fallacies in political debates. Multi-Fusion BERT computes logits ($L$) for each feature by employing a specialized *TokenForClassification* Transformer model adapted to the number of labels: 3 for components and relations, and 17 for part-of-speech tags. The architectures for argumentative features for components and relations share the same parameters, enabling us to obtain *logits* for both components and relations. An additional model, based on the number of PoS

---

[5]The reported results represent the average performance based on the macro average F1 score.

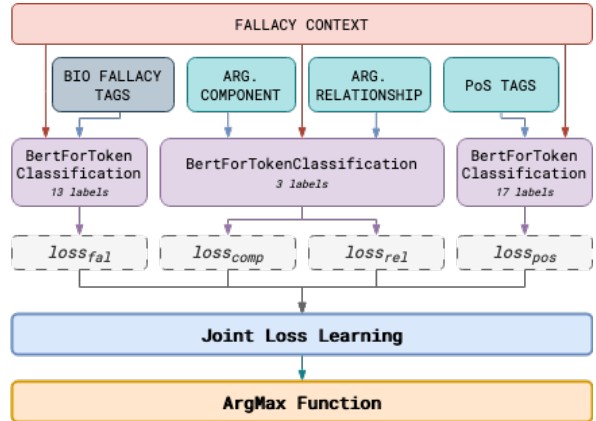

Figure 1: MultiFusion BERT with $joint_{loss}$ approach.

tags (i.e., 17), is used to obtain *logits* for PoS features. Consequently, distinct losses are computed for each model: fallacy loss ($loss_{fal}$), component loss ($loss_{cmp}$), relation loss ($loss_{rel}$), and part-of-speech loss ($loss_{pos}$). These individual losses are combined by multiplying them with an arbitrary $\alpha$ value of 0.1, yielding a unified average loss referred to as the $joint_{loss}$ (Vorakitphan et al., 2021). In our study, we opted for empirically investigating the optimal alpha value that yielded superior performance, as evidenced by our experiments (see Appendix D for the exhaustive evaluation). The back-propagation function incorporates all losses in the following way: $joint_{loss} = \alpha * \frac{(loss_{fal}+loss_{cmp}+loss_{rel}+loss_{PoS})}{N_{loss}}$, where $N_{loss}$ denotes the number of losses considered by the model. We conducted an exploration of various values for the $\alpha$ parameter.

## 4.2 Experimental Setup

All models have been fine-tuned using the ElecDeb60to20 dataset. The implementation was based on HuggingFace[6] version 4.30. and on PyTorch 1.7.0. All models utilize the Adam optimizer, with a gradient clipping set to 10, a dropout of 0.1, a learning rate of $4e-05$ and a training batch size of 8 and a test batch size of 4. The training process consists of 4 epochs, during which the models are fine-tuned and optimized. The dataset was split with 90% for training and 10% for testing, ensuring balanced distribution on the fallacy labels using stratification. The partitioning was performed using the train_and_test_split function from the

scikit-learn[7] library. The random seed was set to 42. The PoS tags were obtained using the spaCy[8] library. The maximum tensor size is set to 256, ensuring that all necessary text information is included without truncation. The representation of this encoding is showed in the Appendix A.

The proposed neural architecture contains 328 million parameters[9]. This large parameter count enables the model to capture intricate patterns and dependencies within the data, enhancing its capacity for complex information processing and generating more accurate predictions. We evaluated the "MultiFusion" approach on the baseline model that showed the best results. Despite having approximately half of the trainable parameters of BERT (e.g., 65M for DistilBERT vs. 109M for BERT), BERT outperformed DistilBERT. Consequently, we adopted BERT as the architecture for implementing the approach described above. We utilized the Nvidia Quadro RTX 8000 GPU (32 GB) for our experiments. The average runtime was of 21 minutes for training and testing all the configurations of our models.

## 5 Evaluation

Table 3 presents the results of the tested models for fallacies detection in the political debates. Results are calculated using the macro average F1 metric, considering the following fallacy labels: *(i)* Ad Hominem, *(ii)* Appeal to Authority, *(iii)* Appeal to Emotion, *(iv)* False Cause, *(v)* Slippery Slope, *(vi)* Slogans, and *(vii)* Other (*B* and *I* labels are merged). Despite the relatively smaller size of the dataset and the task complexity, the results obtained from the different models are promising. As introduced before, among the baselines, BERT "*db-mdz/ bert-large-cased-finetuned-conll03-english*" achieved the best performance. Thus, MultiFusion BERT, incorporating argumentative features (components and relations) as well as PoS tags, significantly outperformed the other models (the performance increase with respect to BertFTC is of 2.12%).

---

[6]https://huggingface.co/docs/transformers/index

[7]https://scikit-learn.org/stable/modules/generated/sklearn.model_selection.train_test_split.html
[8]https://spacy.io
[9]The standard BertFTC model is around three times smaller, approximately 109 million parameters.

## 5.1 Ablation Tests

To better analyze the impact of the different features incorporated in our architecture, we carried out ablation tests. Table 4 presents the results obtained by MultiFusion BERT using all possible combinations of *(i)* argumentative components, *(ii)* argumentative relations, and *(iii)* context PoS tags. Incorporating argumentative components, relations, and PoS features individually or in pairs resulted in a decline in performance compared to the best baseline results (i.e., BertFTC "*dbmdz/ bert-large-cased-finetuned-conll03-eng.*"), with an average degradation of 4.35% across the different configurations (excluding the one considering all three features). In contrast, when all three features are included (as described in Section 4.1.2) a significant improvement in model performance is observed, highlighting the importance of considering all of them together for fallacy detection.

## 5.2 Error Analysis

| Label | precision | recall | f1-score | support |
|---|---|---|---|---|
| AdHominem | 0.99 | 0.77 | 0.87 | 739 |
| AppealtoAuthority | 0.90 | 0.78 | 0.83 | 1'049 |
| AppealtoEmotion | 0.82 | 0.77 | 0.79 | 2'224 |
| FalseCause | 0.82 | 0.86 | 0.84 | 321 |
| Slipperyslope | 0.90 | 0.88 | 0.89 | 332 |
| Slogans | 0.00 | 0.00 | 0.00 | 49 |
| O | 0.90 | 0.95 | 0.93 | 7'914 |
| accuracy | | | 0.89 | 12'628 |
| macro avg | 0.76 | 0.72 | **0.74** | 12'628 |
| weighted avg | 0.89 | 0.89 | 0.89 | 12'628 |

Table 5: Classification report of Fallacy Detection and Classification with *B* and *I* labels merged.

Table 5 provides an in-depth analysis of MultiFusion BERT's performances on the test set, considering the different target labels[10]. Notably, the identification of tokens labeled as *Slogans* exhibits the poorest results, despite being relatively easier to recognize for humans. This can be due to the limited presence of examples/tokens in both the training and the test set[11]. In addition, these results point out that recognizing slogans within political debates involves factors beyond syntactic and argumentative features (mostly semantics and pragmatics). On the contrary, tokens labeled as "Slippery Slope" and "False Cause" (with 332 and 321 examples, respectively) are much better classified

[10]A detailed analysis of the performances with BIO tokens is provided in Appendix B

[11]A detailed table with the count of each token can be found in Appendix C

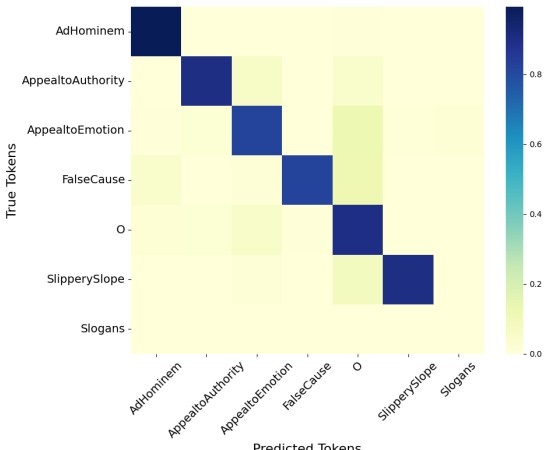

Figure 2: Normalized confusion matrix MultiFusion BERT. BIO labels are merged. Normalization is performed using the number of true elements in each class.

by the model, showing the highest performances (0.89 and 0.84). The definition of "Slippery Slope" revolves around portraying improbable or exaggerated consequences arising from a specific action, and argumentative components are often used to the cause, as well as semantic nuances well captured by the model.

The results obtained for the other labels are in line with those in (Goffredo et al., 2022) for the classification task only. The addition of new fallacious examples from the 2020 debates kept unchanged the distribution of fallacies with respect to the previous debates, suggesting that the detection of fallacious snippets remains consistent and stable across different debate contexts.

For a better understanding of the predictions made by the MultiFusion BERT we analyze the normalized confusion matrix visualized in Figure 2. The normalized version is preferred due to the dataset class imbalance. Notably, although the class with the highest F1 score is *O* (representing non-fallacies), the confusion matrix reveals that the model tends to over-predict instances in this category. As observed in the column of the predicted *O* class, false positives are the most prevalent in the non-fallacious tokens. Moreover, *False Cause* and *Appeal to Emotion* are the classes that the models misinterpret the most as non fallacious. In a smaller proportion, the model misclassifies instances of *Appeal to Authority* as *Appeal to Emotion*.

Table 6 shows a few misclassified fallacy snippets. In the first example, the argument is misclassified as Appeal to Emotion instead of Appel to Authority, because the model is misled by the word

| Fallacy snippet | True fallacy | Pred. fallacy |
|---|---|---|
| **Franklin Roosevelt said in 1932 that the only thing we have to fear is fear itself.** | Appeal to Authority | **Appeal to Emotion** |
| **As the President said the other night, there will always be troubles in this ol' world, but the United States of America** *can be counted on to provide the vision that the world looks for from the United States of America.* | O | **Appeal to Authority** and *O* |
| **But as Admiral Yarnell has said, and he's been supported by most military authority, these islands that we're now talking about are not worth the bones of a single American soldier;** *and I know how difficult it is to sustain troops close to the shore under artillery bombardment.* | Appeal to Authority | **Appeal to Emotion** and *O* |
| **In a place like Chicago, where thousands of people have been killed,** *thousands over the last number of years, in fact, almost 4,000 have been killed since Barack Obama became president, overall almost 4,000 people in Chicago have been killed. We have to bring back law and order.* | False Cause | **False Cause** and *O* |

Table 6: Examples of misclassification using the best-performing model. Underlined text is to highlight the true label for each token, whereas **Bold** is for the predicted fallacy and *Italic* for predicted *O* tokens.

"fear," which carries an important emotional connotation. In the next example, the argument is erroneously classified as being an Appeal to Authority argument whilst it is not a fallacious argument (O). The third example shows another instance where the model confuses Appeal to Authority with Appeal to Emotion, while also failing to identify part of the fallacy in general. The third and the fourth examples show where the model partially identifies the correct fallacy or its absence.

## 6 Conclusion

Existing argumentation schemes (Walton, 1995) to identify flawed and invalid forms of reasoning often fall short when applied to fallacious arguments employed in real-world contexts like political debates. To tackle this challenge, the contribution of this paper is twofold. First, we extended the ElecDeb60to16 dataset by incorporating the Trump vs. Biden 2020 presidential debate along with argumentative annotations and fallacies. Second, we proposed and evaluated MultiFusion BERT, a transformer-based architecture that combines the debate text, the argumentative features (i.e., components and relations), and engineered features to perform the fallacy detection and classification task. Our results highlight the main role of argumentative features in the correct identification and classification of fallacious arguments. This approach yields an average performance improvement of 2.12% compared to baseline methods and competing approaches.

As future research, we intend to delve deeper into fallacious argumentation by integrating knowledge in order to address more challenging fallacy categories like causal ones, where reasoning and knowledge-based features are required to identify the fallacy. Our further objective is to generate valid arguments from identified fallacious ones and their context. Additionally, a challenge we aim to tackle is to explore ways to counter the formal invalidity of fallacious arguments through the generation of new arguments.

## 7 Limitations

Some limitations of this work require a discussion. Firstly, the used training corpus is focused on US political debates, which restricts the applicability of the model to English-language contexts only. Furthermore, the imbalanced distribution of labels had a noticeable impact on the model's performance and its ability to generalize during prediction. For instance, the label "Slogan" was significantly underrepresented compared to other labels, further affecting the model's performance. Finally, it is important to consider that the GPU requirements, specifically the need for Nvidia RTX 8000 with 32GB VRAM, may present limitations on the practical utilization of these models in resource-constrained environments. These limitations highlight the need for further research to address the dataset limitations with respect to the employed language and the label balance, to improve the model architecture, and explore additional strategies to enhance fallacy detection through knowledge injection.

## 8 Acknowledgments

This work was partly supported by the French government, through the 3IA Côte d'Azur Investments in the Future project managed by the National Research Agency (ANR) with the reference number ANR-19-P3IA-0002. This work was partly supported also by EU Horizon 2020 project AI4Media, under contract no. 951911 (https://ai4media.eu/). This work has been partially supported by the ANR project ATTENTION (ANR-21-CE23-0037).

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

## A    Encoding

A representation of the dataset's encoded example is illustrated in the Figure 3, demonstrating how it is prepared for input into the architecture for token label prediction. Notably, the argument features align with the offset mapping approach employed by the tokenizer during tokenization.

Each argumentative feature is represented by a tensor of length 256 (the maximum token length of 216 is used) filled with label IDs (0: *None*, 1: *Claim/Support*, 2: *Premise/Attack*) up to the maximum length. The same process is applied to the Part-of-Speech tensor, where each tag is converted into its corresponding ID (0: *ADJ*, 1: *ADP*, 2: *ADV*, 3: *AUX*, 4: *CCONJ*, 5: *DET*, 6: *INTJ*, 7: *NOUN*, 8: *NUM*, 9: *PART*, 10: *PRON*, 11: *PROPN*, 12: *PUNCT*, 13: *SCONJ*, 14: *SYM*, 15: *VERB*, 16: *X*).

## B    Detailed Model Performance

In this appendix section, a comprehensive classification report of the best-performing model is presented, considering all BIO labels and all the three features (argumentative components, argumentative relations and PoS tags). This report provides a thorough assessment of the model performance, offering valuable insights into its accuracy and effectiveness in recognizing and classifying different categories to every token.

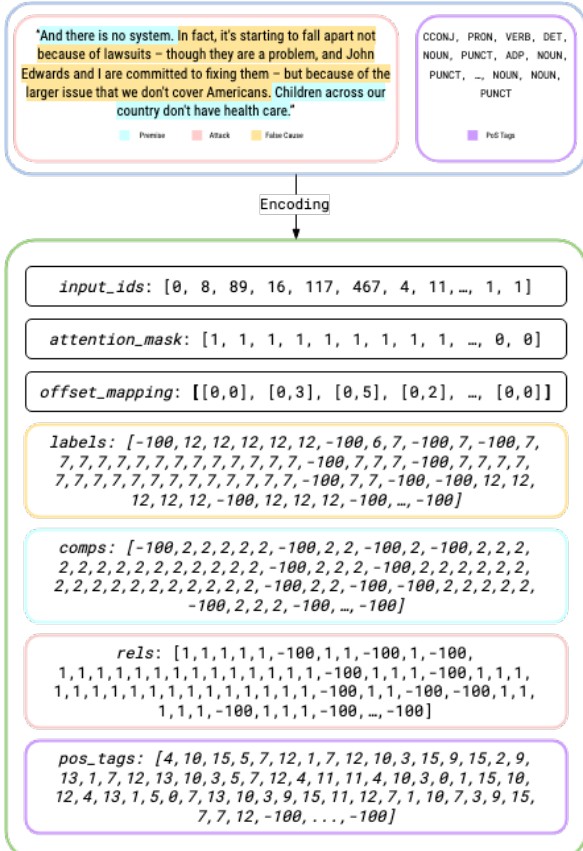

Figure 3: Encoded example of a single item of the dataset ElecDeb60to20.

| Label | precision | recall | f1-score | support |
|---|---|---|---|---|
| B-AdHominem | 1.00 | 0.19 | 0.31 | 27 |
| B-AppealtoAuthority | 0.75 | 0.50 | 0.60 | 30 |
| B-AppealtoEmotion | 0.72 | 0.39 | 0.51 | 120 |
| B-FalseCause | 0.75 | 0.33 | 0.46 | 9 |
| B-Slipperyslope | 0.33 | 0.12 | 0.18 | 8 |
| B-Slogans | 0.00 | 0.00 | 0.00 | 5 |
| I-AdHominem | 0.98 | 0.79 | 0.88 | 712 |
| I-AppealtoAuthority | 0.90 | 0.78 | 0.84 | 1'019 |
| I-AppealtoEmotion | 0.81 | 0.78 | 0.79 | 2'104 |
| I-FalseCause | 0.81 | 0.87 | 0.84 | 312 |
| I-Slipperyslope | 0.89 | 0.89 | 0.89 | 324 |
| I-Slogans | 0.00 | 0.00 | 0.00 | 44 |
| O | 0.90 | 0.95 | 0.93 | 7'914 |
| accuracy | | | 0.88 | 12'628 |
| macro avg | 0.68 | 0.51 | **0.56** | 12'628 |
| weighted avg | 0.88 | 0.88 | 0.88 | 12'628 |

Table 7: Classification report of fallacy entity classification with BIO labels.

## C    Token Distribution

Table 8 shows the distribution of the BIO tokens among the training and test set.

## D    Analysis of Different $\alpha$ Values

MultiFusion BERT's loss function is defined as

$$joint_{loss} = \alpha * \frac{(loss_{fal}+loss_{cmp}+loss_{rel}+loss_{PoS})}{N_{loss}},$$

| Label | Train | Test | Total |
|---|---|---|---|
| B-AdHominem | 243 | 27 | 270 |
| B-AppealtoAuthority | 272 | 30 | 302 |
| B-AppealtoEmotion | 1'073 | 120 | 1'193 |
| B-FalseCause | 84 | 9 | 93 |
| B-Slipperyslope | 71 | 8 | 79 |
| B-Slogans | 47 | 5 | 52 |
| I-AdHominem | 4'855 | 712 | 5'567 |
| I-AppealtoAuthority | 8'707 | 1'019 | 9'726 |
| I-AppealtoEmotion | 17'408 | 2'104 | 19'512 |
| I-FalseCause | 3'076 | 312 | 3'388 |
| I-Slipperyslope | 2'487 | 324 | 2'811 |
| I-Slogans | 254 | 44 | 298 |
| O | 74'493 | 7'914 | 82407 |
| **Total** | **113'070** | **12'628** | **125'698** |

Table 8: Distribution of BIO fallacy tags among the train and test set.

| $\alpha$ | Features | | | Avg macro |
|---|---|---|---|---|
| | Components | Relationships | PoS | F1 Score |
| | ✓ | | | 0.6922 |
| | | ✓ | | 0.6922 |
| | | | ✓ | 0.7212 |
| 0.1 | ✓ | ✓ | | 0.7278 |
| | ✓ | | ✓ | 0.7166 |
| | | ✓ | ✓ | 0.7166 |
| | ✓ | ✓ | ✓ | **0.7394** |
| | ✓ | | | 0.7054 |
| | | ✓ | | 0.7054 |
| | | | ✓ | **0.7214** |
| 0.3 | ✓ | ✓ | | 0.6889 |
| | ✓ | | ✓ | 0.7160 |
| | | ✓ | ✓ | 0.7160 |
| | ✓ | ✓ | ✓ | 0.7084 |
| | ✓ | | | 0.7057 |
| | | ✓ | | 0.7057 |
| | | | ✓ | 0.6817 |
| 0.5 | ✓ | ✓ | | **0.7366** |
| | ✓ | | ✓ | 0.7054 |
| | | ✓ | ✓ | 0.7054 |
| | ✓ | ✓ | ✓ | 0.7070 |

Table 9: MultiFusion BERT's average macro F1 scores for fallacy detection using different features and values of the $\alpha$ parameter. The scores are based on an average of 3 runs. *B* and *I* labels were merged.

where $N_{loss}$ denotes the number of losses considered by the model. We conducted an exploration of various values for the $\alpha$ parameter. Table 9 shows that the impact of the parameter varies depending on the features used in the model. That is, none of the values of $\alpha$ yields a generalized improvement in the macro F1 score across all feature combinations.