# OpenReview forum: "Argument-based Detection and Classification of Fallacies in Political Debates"
_EMNLP/2023/Conference — EMNLP 2023 Main_

### Official Review · Reviewer_fe3c · 2023-08-04

**Typos Grammar Style And Presentation Improvements:** NA
**Soundness:** 4

**Excitement:**

4: Strong: This paper deepens the understanding of some phenomenon or lowers the barriers to an existing research direction.

**Missing References:**

NA

**Paper Topic And Main Contributions:**

This paper focuses on the identification and study of fallacies within political debates. It makes a significant contribution by introducing a new dataset, ElecDeb60to20, along with a novel model that could potentially have a far-reaching impact on this subfield.


**Questions For The Authors:**

1. **Alpha Parameter Selection:** What is the rationale behind setting alpha to 0.1? In general, alpha might be considered as a coefficient to assign different weights to various losses (e.g., Loss1 -> 0.5, Loss2 -> 0.4, Loss3 -> 0.1, where the sum is 1). Could you provide more insight into this choice?
2. **Final Target Clarification:** Could you clarify the final target in your classification with respect to Figure 1? Understanding this aspect could provide a clearer picture of the model's objectives and the overall methodology.


**Reasons To Accept:**

- **Relevance to Contemporary Issues:** The task of fallacy detection and classification holds substantial importance in the current landscape of NLP. It's especially significant in contributing to broader subjects like ethics in NLP and the nuanced detection of misinformation.
- **Potential Community Impact:** The introduction of the ElecDeb60to20 dataset is a commendable effort that may act as a catalyst for further research and innovation in the community.
- **Modeling and Experimentation:** The modeling approach and conducted experiments appear robust and compelling. The design illustrated in Figure 1 offers a simplistic yet effective way to fuse three different losses. The use of the BIO (Beginning, Inside, Outside) setting adds to the transparency of the models' decision-making process, reflecting a thoughtful design choice.

**Reasons To Reject:**

While the paper's strengths are many, one area of concern is the lack of detailed information regarding the Institutional Review Board (IRB) permissions for data collection. Clear documentation and adherence to ethical standards are crucial, and the absence of this information might raise questions about the dataset's collection methodology.


**Reproducibility:**

4: Could mostly reproduce the results, but there may be some variation because of sample variance or minor variations in their interpretation of the protocol or method.

**Reviewer Confidence:**

3: Pretty sure, but there's a chance I missed something. Although I have a good feel for this area in general, I did not carefully check the paper's details, e.g., the math, experimental design, or novelty.

---

> ### Author Rebuttal · Authors · 2023-08-28
>
> We thank the reviewer for the insightful comments.
> Concerning the permissions for data collection: we would like to clarify that the debates that constitute the ElecDeb60To20 dataset were sourced from the website of the Commission on Presidential Debates, which openly provides transcripts of debates broadcasted on television and featuring the prominent candidates for presidential and vice-presidential nominations in the United States. All information on this website is accessible to the public. Also, the annotations of this dataset were carried out by annotators with a background in computational linguistics, that followed established ethical guidelines and best practices throughout the annotation process.
>
> Q1. Alpha Parameter Selection: Our methodology draws inspiration from the work of Vorakitphan et al. (2021), who applied this approach with an alpha value of 0.5. In our study, we opted for empirically investigating the optimal alpha value that yielded superior performance, as evidenced by our experiments (elaborated upon comprehensively in Appendix D).
>
> Q2. The primary objective of our work is to automatically identify segments within texts or debates and classify them into one of six specified labels corresponding to the initially mentioned fallacies. Illustrated in Figure 1 is the model's architecture designed to realize this objective, achieved through the incorporation of various inputs: the BIO tags associated with the fallacies, POS tags, Tags corresponding to the category of argumentative components (Claim, Premise, None), Tags representing the type of argumentative relationship (Attack, Support, None). Subsequently, the loss for each input is computed, and these individual losses are combined into a unified "joint loss" by applying an alpha coefficient to the average of the preceding losses. This approach facilitates the synthesis of multiple input sources into a cohesive training objective.

---

### Official Review · Reviewer_YLEV · 2023-08-04

**Soundness:** 4

**Excitement:**

3: Ambivalent: It has merits (e.g., it reports state-of-the-art results, the idea is nice), but there are key weaknesses (e.g., it describes incremental work), and it can significantly benefit from another round of revision. However, I won't object to accepting it if my co-reviewers champion it.

**Paper Topic And Main Contributions:**

This paper focuses on the extension of an existing corpus of U.S. political debates, covering English transcripts from 1960 to 2016, by extending it with additional annotations of argument components, relations, and a unique classification of six fallacy categories, comprised of 2357 features. Another key contribution is the development of a novel methodology named MultiFusion. This approach leverages transformer models to detect fallacious text snippets by integrating the learning of argument components, part-of-speech tags, relationships, and fallacy tags. The proposed method has demonstrated robust performance with an f1 macro score of .74. In totality, the paper is well-written, easy to follow, and methodically structured, presenting a solid new layer of information that contributes to the research community.

**Questions For The Authors:**

Regarding the limitations, you mentioned the aspect of the smaller model, which I also think is very important. I wondered if a smaller model like "distilbert" enriched with the presented additional features in joint training could also give good results. Have you also tested your ModelFusion with smaller models?

**Reasons To Accept:**

* The authors provide a solid motivation for their task of fallacy detection at the token level within the context of political debates.
* The most crucial aspect of the paper is the extension of a relevant dataset with fine-grained additional information at the token-span level. This extensive data augmentation enhances the study's comprehensiveness and allows for more nuanced explorations. It also broadens the scope of potential applications, fostering an environment for future research and advancements in the field.
* The authors have proposed a novel approach, compared to multiple baselines, to execute the task of fallacy detection. This new method is sound, offering a fresh perspective on addressing such a complex problem. Moreover, introducing the concept of "fusing" additional features is an exciting development and has been shown to enhance the model's performance moderately.

**Reasons To Reject:**

* The study holds the potential to deliver even greater insight if it could allocate more focus toward the field of political research, especially the effects and consequences of such arguments. Adding this focus could make the discussion richer and wider.
* The new method mentioned in the paper (ModelFusion) could be discussed in more detail. Understanding the rationale behind the authors' selection of features such as PoS over other (meta-)aspects could add another layer of depth to this work. It appears that a grounded motivation, in terms of political debates and similar features, is somewhat lacking in the paper. Currently, the paper is strong on its focus on NLP aspects. Adding more context on how these decisions relate to the domain of political debates etc. could make it even more comprehensive.

**Reproducibility:**

1: Could not reproduce the results here no matter how hard they tried.

**Reviewer Confidence:**

4: Quite sure. I tried to check the important points carefully. It's unlikely, though conceivable, that I missed something that should affect my ratings.

---

> ### Author Rebuttal · Authors · 2023-08-28
>
> We thank the reviewer for the insightful comments.
> 1) We agree on the importance of exploring fallacies and the impact of such arguments in the field of political research. If our work is accepted at EMNLP, we will dedicate part of the extra-page to add such discussion, which is our current work in collaboration with colleagues studying philosophy of language and political argumentation.
>
> 2) Concerning the features selection (argumentative features and Part-of-Speech):
> i) Argumentative features have been included to improve the model's understanding of an argument's underlying structure, enabling it to detect when this structure is compromised. Since fallacies involve faulty reasoning, we hypothesized that providing this information to the model would be relevant. Results in Table 4 provide empirical evidence to support such hypothesis.
> ii) The integration of  PoS information was driven by the observation that certain fallacies exhibit distinctive grammatical patterns that can be more easily discerned using PoS tagging. For example, in the "loaded language" fallacy (a subcategory of the "appeal to emotion" category)  the intensity of a phrase is often increased by using emotionally loaded phrases, expressed through the use of adjectives and adverbs. Similarly, in the ad hominem fallacy, where the focus shifts from attacking the argument to targeting the character, motives, or personal qualities of the arguer, it is usual to reference someone using a noun or pronoun and subsequently employ adjectives with negative connotations.
>
> 3) Concerning  reproducibility: In Section 4.2 we detail the experimental setup: in particular, we detail the model parameters as well as the libraries version used during the training and evaluation phases. We will make both the dataset and the source code available through a dedicated GitHub repository upon paper acceptance.
>
> 4) Regarding the discussed architectures, we employ transformer-based architectures in both their basic configuration and in a specialized configuration, in both "cased" and "uncased" variations, as well as both standard and "large" configurations. Section 4.1.1  summarizes the  architectures chosen for evaluation, among which the DistilBERT model is included. We decided to evaluate the "MultiFusion" approach on the baseline model that yielded the most favorable outcome. Despite DistilBert having approximately half the number of trainable parameters compared to BERT (65 million for DistilBert vs. 109 million for BERT), the performance of BERT was superior. As a result, we selected BERT as the architecture to which we applied the approach introduced in the paper.

---

### Official Review · Reviewer_CQr9 · 2023-08-05

**Soundness:** 4

**Excitement:**

4: Strong: This paper deepens the understanding of some phenomenon or lowers the barriers to an existing research direction.

**Paper Topic And Main Contributions:**

The paper presents two main contributions. First, it extends an existing resource, the ElecDeb60to20 dataset, which includes annotated U.S. political debates from presidential campaigns (1960-2016) with argument components, relations, and six fallacy categories (Ad Hominem, Appeal to Authority, Appeal to Emotion, False Cause, Slippery Slope, and Slogans). The extension incorporates the 2020 presidential campaign debates (Trump-Biden) with all corresponding annotations. Second, the paper introduces a novel approach using Transformers, named MultiFusion BERT, to detect and classify fallacious text snippets in these debates across the six fallacy categories. By encoding argument components (Premise, Claim), relations (Support, Attack), and PoS tags, the approach successfully identifies and classifies fallacious arguments. Experimental results demonstrate that the proposed method outperforms standard baselines and concurrent approaches, achieving an average F1-score of 0.74. The originality lies in proposing the first neural architecture specifically for detecting and classifying fallacious arguments in political debates, outdoing competing methods.

**Reasons To Accept:**

This paper exhibits a well-organized structure, systematically presenting each aspect of the research. The major claims are not only articulated clearly but are also substantiated by comprehensive experimental results, lending credibility to the findings. The authors introduce a novel model architecture that is both intriguing and relevant to the field of natural language processing (NLP). The innovative design contributes to a marked improvement in classification performance, demonstrating superiority over a wide array of currently popular methods. This comparative analysis strengthens the paper's position within the existing body of work. Furthermore, the authors undertake a meticulous analysis of the model's errors, providing insights into the underlying causes and potential areas for further refinement. They also engage in a thoughtful discussion of the model's limitations, showcasing a balanced perspective. This critical examination not only adds depth to the paper but also guides future research directions. Overall, the paper represents a significant contribution to the field of NLP, with a blend of originality, rigorous experimentation, and thoughtful analysis. It is a commendable piece of research that may inspire further exploration and development in the domain

**Reasons To Reject:**

The proposed paper's primary limitation lies in its reliance on a relatively small dataset for the evaluation of the method. Such a limited scope of data undermines the robustness of the experimental validation and raises concerns about the generalizability of the findings. This lack of comprehensive evaluation decreases the paper's potential contribution to the field and calls for a more thorough investigation before it can be considered for acceptance

**Reproducibility:**

3: Could reproduce the results with some difficulty. The settings of parameters are underspecified or subjectively determined; the training/evaluation data are not widely available.

**Reviewer Confidence:**

4: Quite sure. I tried to check the important points carefully. It's unlikely, though conceivable, that I missed something that should affect my ratings.

---

> ### Author Rebuttal · Authors · 2023-08-28
>
> We thank the reviewer for the insightful comments.
> We agree on the fact that having a bigger dataset would strengthen even more our findings, and this is also why we extended the ElecDeb60To16 dataset by incorporating the most recent Trump-Biden presidential debates. The size of the considered dataset - consisting of approximately 1,600 fallacy examples, is well within the range of existing fallacy corpora (e.g. Musi et al (2022), Sahai et al (2021), Jin et al (2022)), making it a suitable benchmark to prove the robustness of the experimental validation.
> Moreover, the ElecDeb60To16 is the only available dataset - and therefore the largest we could use - for political debates, which has been proven to be the main context in which fallacies are employed to mislead the audience.

---

### Meta-Review · Area_Chair_KL3i · 2023-09-18

**Recommendation:** 4

**Metareview:**

This paper adds argumentation related annotations to an existing dataset on US political debates. No additions include argument component and relation annotations as well as 6 common logical fallacies.  It also presents MultiFusion BERT, a model to identify text snippets with a logical fallacy, achieving .74 F1.

Reviewers appreciated the strong motivation and the resource the authors construct, as well as a novel model. The paper would benefit from a more detailed description of the model with justifications for the design choices. Also, either expanding to wider domains or providing more detailed discussions within the political science domain would be helpful.

---

### Decision · Program_Chairs · 2023-10-07

**Decision:**

Accept-Main

**Comment:**

This paper adds argumentation related annotations to an existing dataset on US political debates. No additions include argument component and relation annotations as well as 6 common logical fallacies.  It also presents MultiFusion BERT, a model to identify text snippets with a logical fallacy, achieving .74 F1.

Reviewers appreciated the strong motivation and the resource the authors construct, as well as a novel model. The paper would benefit from a more detailed description of the model with justifications for the design choices. Also, either expanding to wider domains or providing more detailed discussions within the political science domain would be helpful.